# Single Image De-Raining via Improved Generative Adversarial Nets [note 1]

**DOI:** 10.3390/s20061591

**Published:** 2020-03-12

**Authors:** Yi Ren, Mengzhen Nie, Shichao Li, Chuankun Li

**Affiliations:** 1China Academy of Electronics and Information Technology (CAEIT), Beijing 100041, China; englishhsl@126.com; 2School of Electrical and Information Engineering, Tianjin University, Tianjin 300011, China; nmztju@163.com (M.N.); lisctju@tju.edu.cn (S.L.); 3School of Information and Communication Engineering, North University of China, Taiyuan 030051, China

**Keywords:** image de-raining, generative adversarial network, rain estimation, refinement network

## Abstract

Capturing images under rainy days degrades image visual quality and affects analysis tasks, such as object detection and classification. Therefore, image de-raining has attracted a lot of attention in recent years. In this paper, an improved generative adversarial network for single image de-raining is proposed. According to the principles of divide-and-conquer, we divide an image de-raining task into rain locating, rain removing, and detail refining sub-tasks. A multi-stream DenseNet, termed as Rain Estimation Network, is proposed to estimate the rain location map. A Generative Adversarial Network is proposed to remove the rain streaks. A Refinement Network is proposed to refine the details. These three models accomplish rain locating, rain removing, and detail refining sub-tasks, respectively. Experiments on two synthetic datasets and real world images demonstrate that the proposed method outperforms state-of-the-art de-raining studies in both objective and subjective measurements.

## 1. Introduction

Rainy images not only have poor visual qualities, but also heavily affect analysis related tasks, such as detection, classification, recognition, and tracking. Therefore, developing algorithms for automatically removing rain streaks is essential.

Many studies have been proposed to solve the image de-raining problem. The studies can be categorized as prior-based methods, convolutional neural network (CNN)-based methods and Generative Adversarial Networks (GAN)-based methods. The prior-based methods achieve various degrees of success [1,2,3,4]. However, the prior information is usually insufficient to cover all shapes, densities or directions of rain streaks. Therefore, prior-based methods may only work in a part of rain conditions. Furthermore, most of the existing prior-based rain removal studies pay attention to removing rain streaks rather than restoring the information of input images. The result is that they usually have blurry outputs. In past years, CNN-based methods have shown promising results in many computer vision tasks including single image de-raining tasks [5,6,7,8]. However, the problem is that they usually lose detail information, especially the texture which looks like rain streaks. One reason is that they consider the image de-raining as a signal separation task. However, the pixels of rain image are superposition of rain streaks and pixels of rain-free background image, and detail restoring should be considered to be as important as removing rain streaks. The Generative Adversarial Networks [9] which are originally used in image restoration task can generate realistic images. Recent years, several GAN-based de-raining methods have been proposed such as ID-CGAN [10], PAN [11] and FS-GAN [12]. All aforementioned works use a single model to solve the de-raining problem. However, it is difficult to solve the unconstrained de-raining problem using a single model. We observe that decomposing the image de-raining task into several simple sub-tasks, such as rain location, rain removal, and detail refinement can improve the de-raining performance.

In this paper, we propose a GAN-based de-raining method called IGAN-SID, which is an extension of our previous work GAN-SID [13]. The major contributions of this paper can be summarized as follows.

In contrast to previous CNN (GAN)-based de-raining methods, the proposed model uses the divide-and-conquer strategy. The de-raining task is decomposed into rain estimating, rain removing, and detail refining sub-tasks.The rain estimation network (REN) is proposed to generate a rain location map to indicate whether rain streaks exist or not at given location. The estimated rain location map can guide the followed de-raining generative network to generate better de-raining result.GAN is proposed to de-rain the rainy image. Squeeze-and-Exciation (SE) modules are used in generative network. Instance normalization (IN) [14] is used to normalize the features. The trainable perceptual loss function is used to guide the network to generate rain-free image which is close to real-world rain-free image semantically.A refinement network (RFN) to refine the de-rained image and restore more details is proposed. The input of RFN is the concatenation of the output of the generator and the original rainy image. In this way, RFN can recover the details lost in the generator.The ablation study demonstrates that the proposed REN and RFN greatly improves the de-raining performance with quite a few increases in parameters. The experimental results on two synthetic datasets and real-world images show that the proposed method outperforms state-of-the-art de-raining methods. Figure 1 presents an example of the de-raining result. It can be observed that compared with GAN-SID [13] (state-of-the-art work), the proposed IGAN-SID can remove the rain streaks and enhance details.

## 2. Related Work

In this section, we briefly review related studies on single image de-raining task. The existing studies can be categorized as prior-based, CNN-based and GAN-based methods.

### 2.1. Prior-Based Methods

Kim et al. [1] detect rain streak regions by analyzing the rotation angle and the aspect ratio of the elliptical kernel at each pixel location, and then perform the nonlocal means filtering to remove the rain streaks. Zhu et al. [2] remove rain streaks by decomposing the input image into a rain-free background layer B and a rain streak layer R. Then a joint optimization process is used to remove rain streaks. Zhang et al. [3] learn a set of generic sparsity-based and low-rank representation-based convolutional filters for efficiently representing background rain-free image and rain streaks, respectively. Li et al. [4] uses patch-based priors which are based on Gaussian mixture models for both the background and rain layers to remove the rain streaks. However, the prior-based methods usually cause over-smoothness of the background.

### 2.2. CNN-Based Methods

Recently, deep learning-based methods have shown promising performance. Fu et al. [5] propose DerainNet which combines domain knowledge of image processing and convolutional neural networks. Input rainy images are decomposed as detail layer (the part of high frequency) and base layer (the part of low frequency) by a low-pass filter. The detail layer is processed by a non-linear function (CNN) to erase rain streaks. The base layer is processed by an enhancement method [15] to get a clearer background. The de-rained image is the sum of detail layer and base layer. In another work by the same authors [7], the detail layer is processed by a deep detail network (DDN) to output a negative residual map. The de-rained output image is the sum of negative residual map and input image. Zhang et al. [6] propose a multi-stream dense network (DID-MDN) which leverages features from the different scales. In contrast to other studies, DID-MDN can automatically estimate the rain-density label. The de-raining process is accomplished under the assistance of predicted label. Yang et al. [8] propose a multi-task network called Joint Rain Detection and Removal (JORDER) to learn the binary rain streak map, the appearance of rain streaks, and the clean background. Ren et al. [16] propose a network called progressive recurrent network (PReNet). The PReNet takes advantage of recursive computation by repeatedly unfolding a shallow ResNet. And a recurrent layer is introduced to exploit the dependencies of deep features across stages. Wang et al. [17] proposes a spatial attentive network (SPANet) to remove rain streaks in a local-to-global manner. SPANet uses a two-round four-directional IRNN architecture to generate an attention map. The attention map can guide following networks remove rain streaks via the learned negative residuals.

### 2.3. GAN-Based Work

GAN is first proposed in [9]. It was used to generate samples which distribution similar to real-world samples [18]. GAN achieves promising performance in image-to-image translation tasks [19,20,21,22] including super-resolution, semantic segmentation, image in-painting. Essentially, these tasks are learning a mapping from one image to another. The pix2pix-cGAN [19] can achieve paired image translation (e.g., labels to street scene, aerial to map, day to night, edges to photo, etc.), the CycleGAN [20] can achieve unpaired image translation (e.g., Monet to photos, zebras to horses, summer to winter, etc.). The StarGAN [21] can achieve multi-domain image translation (changing hair color, age, gender, etc.). Generally, image de-raining can be taken as an image translation task, i.e., mapping the rainy images to rain-free images. Thus, several GAN-based image de-raining methods are proposed. For example, ID-CGAN [10] proposes a GAN-based network which using skip-connection and DenseNet Block. ID-CGAN uses combination of per-pixel loss and perceptual loss to ensure the performance. PAN [11] employs the trainable hidden layers of the discriminative network of GAN to measure the perceptual adversarial loss between de-rained images and corresponding rain-free images. FS-GAN [12] trains a GAN model for which the supervision from ground truth is imposed on different layers of the generator network.

As discussed, the studies which mentioned above usually cause blurry results due to neglecting of detail refinement. In our work, the proposed model uses a GAN model to remove rain streaks. In addition, two auxiliary models are used to locating the rain streaks and refining the details.

## 3. IGAN-SID Network

In this section, we explain the architecture and loss functions of the proposed model. The overview of the proposed model is illustrated in Figure 2. As shown in the figure, the proposed model consists of three parts: REN, GAN, and RFN. The REN aims to estimate the distribution of rain streaks from input image. The GAN aims to remove the rain streaks from input image. In addition, the RFN aims to refine the detail information from de-rained image.

### 3.1. The Rain Estimation Network

As discussed in the previous section, although some of the previous methods achieve impressive performance on image de-raining task, they often tend to get a blurry result or rain residue. One reason is that they usually take rain streaks locating, removing and detail refining as a single task which is challenging for a single model. To solve this problem, DID-MDN [6] using a classifier to classify the rainfall. To train the classifier, DID-MDN develops a new dataset which has different rain labels. In this work, we consider that estimating the rain streaks map can be more directly. Thus, the first step of proposed model is locating the rain streaks. REN as a pre-processing of proposed model is proposed to provide a rain location map. The estimated map can indicate whether a pixel belongs to a rain streak or not. As shown in Figure 3a, given the input image, REN estimates the distribution of rain streaks and outputs a rain location map. The REN uses three-stream dense network which is similar to [6]. The kernel size of each stream is 3×3, 5×5 and 7×7, respectively. Using different kernel size aims to estimate the rain streaks in different spatial receptive fields. Each stream consists of six dense blocks. As an example, the architecture of 5×5 stream is shown in Figure 3b. To reduce the computation complexity, pairs of transition-down and transition-up layers are adopted in 5×5 and 7×7 streams respectively. Each stream generates a feature map of rain streaks. Then three feature maps are concatenated in channel dimension. After a Conv−InstanceNorm−Tanh processing, a rain location map is generated.

### 3.2. The Generative Adversarial Network

As shown in Figure 4, the proposed GAN model consists of two parts: the generative network and the discriminative network. The input of generative network is the concatenation of input image and estimated rain map. The output is de-rained image. To learn the distribution of real-world rain-free images, the input of discriminative network is de-rained image or rain-free image, and the output is the predicted label (real or fake). The architecture of generator consists of downsampling, transforming and upsampling. The discriminator employs an architecture of convolutional neural network.

In generator, the part of transforming is made up of residual blocks. Each residual block uses a Squeeze-and-Excitation (SE) module [24] which can adaptively recalibrate channel-wise feature responses by explicitly modeling interdependencies between channels. The SENet has shown impressive results on classification tasks. It plays a role in single image de-raining task by modeling the importance of feature channels. The experimental results show that SE-ResBlock can achieve superior performance compare with ResBlock.

As shown in Figure 4, the architecture of proposed Generator consists of three parts: first, it has three convolution layers with (7,1,64), (4,2,128) and (4,2,256) for extracting and down-sampling the features, where (7,1,64) represents a convolution layer with kernel size 7, stride 1 and output channels 64. Then 9 SE-ResBlocks are applied to transforming the feature map. In addition, the mirrored deconvolution layers which are corresponding to first three convolution layers are applied for up-sampling the feature map. Finally, a tanh activation layer is applied to generate the rain-free image. For the discriminator, it consists of nine Conv−InstanceNorm−LeakyReLU layers. The architecture of discriminator follows [11].

### 3.3. The Refinement Network

The REN and GAN models can accomplish locating and removing rain streaks tasks. Although the de-rained images from the generator show impressive performance in [13], the rain residue still can be found in heavy rain conditions, as shown in Figure 5. Previous studies usually ignore the refinement or only have a coarse refinement part with two convolution layers [6]. Thus, RFN is proposed to restore the image details. As shown in Figure 6, it consists of skip, body, and head parts. The de-rained image of generator and rainy input image are concatenated as the input of RFN. The reason is that after the processing of generator, some origin details have been lost. The refinement part cannot restore the lost details without any information. To reduce the computation overhead, a stride-convolution layer is applied to down-sample the image. The head part is a convolution layer with IN and ReLU. The body part is a series of improved residual blocks [25]. A residual block can be written as y=f(x)+x. If *x* is optimal, the network only needs to fit f(x)=0 which much easier than fitting f(x)=x. Followed by residual blocks, a pixelshuffle [26] layer is adopted. The skip part consists of a convolution layer and a pixelshuffle layer. The output of skip and body are concatenated in channel dimension. Then a *Tanh* activation function is applied to map the output to [−1,+1]. The architecture is adopted from [25].

### 3.4. The Objective Function

To guide each sub-network to accomplish their task, combined loss functions are used in the proposed model. The loss functions consist of GAN loss, perceptual loss and per-pixel loss.

#### 3.4.1. GAN Loss

GAN loss function gives adversarial goals of generator and discriminator. Specifically, the generator aims to generate samples which is similar to real-world rain-free images. The discriminator aims to judge the input images which come from generator or real world. The generative adversarial objective function for proposed model can be expressed as:(1)$$\begin{array}{c} \min_{G}\max_{D}E_{C∼P_{clear}}\left[\log\left(D\left(C\right)\right)\right]+E_{R∼P_{rain}}\left[\log\left(1-D\left(G\left(R\right)\right)\right)\right] \end{array}$$
where *C* is the rain-free images which come from real-world, *R* means the samples from the pool of synthetic rainy images, *G* means the generator, *D* means the discriminator. G(R) represents the de-rained image of generator.

#### 3.4.2. Perceptual Loss

Perceptual loss aims to explore the discrepancy between real-world images and de-rained images in high-dimensional representations. The representations are extracted from CNNs. The perceptual loss can push thede-rained image similar to real-world rain-free image semantically.

The perceptual loss usually uses a pre-trained model such as VGG [27], InceptionNet [28] or ResNet [29], etc. In the proposed model, we use a trainable hidden-layers to measure the perceptual loss. The layers come from discriminator. The perceptual adversarial loss can be expressed as: (2)$$L_{P_{G}}\left(R,C\right)=\sum_{i=1}^{N}\lambda_{i}P_{i}\left(G\left(R\right),C\right)$$
(3)$$L_{P_{D}}\left(R,C\right)=\max\left(0,m-\sum_{i=1}^{N}\lambda_{i}P_{i}\left(G\left(R\right),C\right)\right)$$
where $P_{i}\left(\cdot ,\cdot \right)$ represents the feature discrepancy which are extracted from *i*-th hidden layer of the discriminative network *D* between the real-world images and de-rained images, ${\left\{\lambda_{i}\right\}}_{i=1}^{N}$ are hyper-parameters which represent the weights of different feature discrepancy. *m* means the margin of calculated discrepancy. When P(G(R),C)≥m, the proposed $L_{P_{D}}$ will have zero gradients and do not affect the generator. It will work at the beginning of training. In practice, the value of *m* is set to 2, *N* is set to 4, $\lambda_{1},\lambda_{2},\lambda_{3},\lambda_{4}$ are set to 5, 1.5, 1.5, 5. The feature discrepancy is calculated as L1 distance.It can be expressed as:(4)$$\begin{array}{c} P_{i}\left(G\left(R\right),C\right)=\left\|H_{i}\left(C\right)-H_{i}\left(G\left(R\right)\right)\right\| \end{array}$$
where $H_{i}\left(\cdot \right)$ is the high-dimension features on the *i*-th hidden layer of the discriminative network *D*.

In the proposed REN model, a pre-trained VGG network [27] is employed to calculate the perceptual loss. The computed result comes from the layer relu1_2 of the VGG-16 model. It can be expressed as:(5)$$L_{P_{R}}=\frac{1}{CWH}\sum_{c=1}^{C}\sum_{w=1}^{W}\sum_{c=h}^{H}\left\|V{\left(r\right)}^{c,w,h}-V{\left(gt\right)}^{c,w,h}\right\|$$
where gt is the ground truth of rain map. It can be gotten from rain-free image and rain image:(6)$$gt=\frac{1}{CWH}\sum_{c=1}^{C}\sum_{w=1}^{W}\sum_{c=h}^{H}\left\|Rain^{c,w,h}-Clear^{c,w,h}\right\|$$

#### 3.4.3. Per-pixel Loss

To accelerate the convergence, Mean-Square Error(MSE) loss is applied in GAN model as per-pixel loss. In addition, MSE loss is also used in REN and RFN. Given a rainy image *I* and generated image *O* with channels *C*, width *W* and height *H*, the MSE loss function can be described as L2 distance:(7)$$L_{MSE}=\frac{1}{CWH}\sum_{c=1}^{C}\sum_{w=1}^{W}\sum_{h=1}^{H}\left\|I^{c,w,h}-O^{c,w,h}\right\|$$

The per-pixel loss and perceptual are similar in formulation. The difference is that the perceptual loss is calculated with feature maps which extracted by CNN and the per-pixel loss is calculated with original image.

#### 3.4.4. Total Loss

In general, the loss function of GAN model can be expressed as:(8)$$\begin{array}{cc} L_{G}\left(R,C\right) & =L_{GAN}+L_{P_{G}}\left(R,C\right)+\lambda_{MSE_{GAN}}L_{MSE}\left(R,C\right) \end{array}$$
(9)$$L_{D}\left(R,C\right)=-\log\left(D\left(C\right)\right)-\log\left(1-D\left(G\left(R\right)\right)\right)+L_{P_{D}}\left(R,C\right)$$

The loss function of REN can be expressed as:(10)$$L_{REN}=\lambda_{MSE_{REN}}L_{MSE}\left(r,gt\right)+L_{P_{R}}$$

The loss function of RFN can be expressed as:(11)$$L_{RFN}=L_{MSE}\left(refine,clear\right)$$
where $L_{GAN}=\log\left(1-D\left(G\left(R\right)\right)\right)$ is generative adversarial loss of GAN. $\lambda_{MSE_{GAN}}$ is set to 10.$\lambda_{MSE_{REN}}$ is set to 5.

## 4. Experiments

In this section, we explain the experimental details and evaluation results on synthetic and real-world datasets. Peak signal-to-noise ratio (PSNR) and structural similarity (SSIM) [30] are used as performance metrics on synthetic datasets. It has been proved that no-inference metrics, for example, NIQE [31], BLIINDS-II [32] and SSEQ [33] do not align well with the human perception quality [34]. One reason is that the metrics are designed to evaluate the quality of an image but not to evaluate the similarity between generated and real-world rain-free images. A rainy image may get a high score under this definition. Thus, the performance on real-world rainy images is evaluated visually due to the fact that the ground-truth of real-world rainy image is not available.

### 4.1. Datasets

Two synthetic datasets are carried for experiments. They are *DID-MDN dataset* [6] and *ID-CGAN dataset* [10]. *DID-MDN dataset* contains 12,000 rainy images for training, and the testing set contains 1200 images. There are three levels of rain density in the dataset, i.e., light, medium and heavy. However, the proposed model can directly estimate the distribution of rain streaks, so that these labels are not used in this work. Due to the fact that PAN [11] has no published code, the *ID-CGAN dataset* is used to compare the performance of PAN and ID-CGAN [10] with proposed model. The dataset is smaller and more difficult than *DID-MDN dataset*. It contains 700 images for training and 100 images for testing.

### 4.2. Implementation Details

All experiments are carried on programming language python and deep-learning framework pytorch. The performance is evaluated on MATLAB. The experiments are executed on GPU Nvidia Titan which has 12GB memory, CPU i7-6700 and RAM 16G. To avoid mode collapse, discriminator and generator are updated alternately. Specifically, we update generator 5 times after updating discriminator once. The REN and generator are trained simultaneously. The RFN would not join the training until the half number of total epochs. The reason is that in the first half of training, the de-rained images have poor quality and difficult to refinement. The optimizer of training is Adam solver with first momentum 0.5, second momentum 0.999 and learning rate 0.0002. Batch size is set to 4. For *DID-MDN dataset*, the image resolution is 512×512. For *ID-CGAN dataset*, following [11], the image resolution is resized to 256×256 to compare with PAN and ID-CGAN conveniently. The number of training epochs is set to 20 for *DID-MDN dataset* and 200 for *ID-CGAN dataset*. The learning rate will be decayed linearly to zero over the final 5 epochs for *DID-MDN dataset* and 100 epochs for *ID-CGAN dataset*.

In addition, Batch Normalizaiton (BN) [35] is used in most of the deep-learning based image de-raining models such as [10,11,36] to solve the covariate shift problem [35]. After applying BN operation, the output features of neurons are normalized to mean value 0 and variance value 1 at batch scale. It means that features may not be normalized for a single sample. In other words, the samples will be affected by the others within same mini-batch. This is ill-suited for single image de-raining task. Hence, we replace the BN by IN [14] to solve the covariate shift problem. IN is suitable for image de-raining task due to the fact that it just normalizes data at instance scale. Figure 7 shows a comparison between BN and IN. It can be observed IN outperforms BN, both in visual quality and objective metrics.

### 4.3. Results in Synthetic Datasets

#### 4.3.1. Ablation experiments

Ablation experiments are employed to identify the contribution of each component. The experiments conclude as follows: w/o IN (applying BN in the proposed model), w/o SE (applying ResBlocks rather than SE-ResBlocks in GAN model), w/o PL (without perceptual loss), GAN+REN, GAN+REN+RFN, where w/o means the proposed model without one part and keep the other parts unchanged. The experiments are applied on *DID-MDN dataset*. The average PSNR and SSIM scores are tabulated in Table 1. It can be observed that IN can improve the performance significantly, as mentioned in Character 3. SE module and perceptual loss function have a positive influence to improving the performance of de-raining model. Proposed REN and RFN can further improve the performance, especially RFN. The number of parameters and computational time are tabulated in Table 2. It can be observed that after integrating REN and RFN, the number of parameters only increase 13.80%.

To visually demonstrate the improvements which are obtained by the proposed REN and RFN, Figure 8 show several synthetic and real-world images. It can be observed that REN can estimate the rain streaks effectively. In the area of high brightness, rain streaks cannot be estimated effectively. This is consistent with intuition that the rain streaks have slighter influence in high brightness area than low brightness area. Comparing the details between output of generator and RFN, it can be easily observed that the proposed RFN can remove the rain residue and restore the image details effectively.

#### 4.3.2. Comparison with state-of-the-art methods

To demonstrate the performance of the proposed method, the proposed model is compared with other GAN-based studies on *ID-CGAN dataset*. The quantitative results are reported in Table 3. It can be observed that compared with other studies, the proposed model boosts the PSNR 23.35 (PAN) to 25.84 and SSIM 0.830 (PAN) to 0.871. The reason is three-fold, one is that the proposed model can use both per-pixel information by using MSE loss and high-level semantic information by using perceptual loss. It means the output of proposed model and the ground truth are close in each segment level. Another reason is that the IN used in the proposed model is more beneficial to image de-raining task, as analyzed before. In addition, the proposed REN model can provide an indication for locating the rain streaks. In addition, the proposed RFN can further eliminate the rain residue after generator. Thus, the proposed model outperforms other GAN-based models.

The proposed model is also compared with other state-of-the-art deep learning-based methods as follows: JORDER [8], DDN [7], DID-MDN [6] and PReNet [16]. Quantitative results are tabulated in Table 4. It can be seen that the proposed method can achieve superior performance. The results show that take single image de-raining task as an image-to-image translation task is an idea that worth exploring.

To verify the effectiveness of the proposed method, visual comparison on DID-MDN dataset is listed in Figure 9. From the figure, it can be seen that the proposed method performs better on metrics of SSIM/PSNR. We can also see that the color information is destroyed and dark speckle is got in JORDER [8] due to using Koschmieder model [37]. DDN [7] can remove most of the rain streaks, but rain residue still can be observed in heavy rain condition due to adding the inaccurate negative residual map and input image directly without refinement. Compared with DDN, DID-MDN [6] has some improvements and is able to get a rain-free result by using a predicted label to guide the process of de-raining. However, slight blur still can be observed, since the label is only the approximation of the actual distribution of rain streaks. PReNet [16] can achieve a good performance on medium and light rain conditions, but the phenomenon of blur still exists in heavy rain condition. Our previous work GAN-SID [13] works well at almost all kinds of rain conditions due to replacing the BN by IN in GAN model. Compared with GAN-SID, the proposed IGAN-SID can further improve the quality of de-rained image because it can estimate the actual distribution of rain streaks and refine details by using REN and RFN respectively.

### 4.4. Results on Real-world Images

The proposed model is also evaluated on commonly used real-world images. The comparison is shown in Figure 10. Most of the images come from the studies [6,10]. As discussed before, JORDER [8] loses much color information and generates some darker speckles. DDN [7] loses some meaningful details and overly removes rain streaks. Since it cannot model the difference of rain pixel and the background pixel. The DID-MDN [6] model may remove the objects which context is similar to rain streaks due to losing distribution of rain streaks. PReNet [16] can achieve good performance on synthetic dataset. However, its performance is not satisfying in real-world images. It cannot remove rain streaks completely such as in the first and second row in Figure 10. GAN-SID [13] (proposed model without REN and RFN) performs better at almost all kinds of rain conditions. Compared with GAN-SID, proposed IGAN-SID can further improve the performance. In addition it can repair some contexts, such as the zoomed area in the third row.

## 5. Conclusions

In this paper, we proposed an Improved Generative Adversarial Network for Single Image De-raining (IGAN-SID). Instead of modeling the de-raining task with a single network, we divide the de-raining task into several specific sub-tasks, i.e., rain estimation, rain removal, and detail refinement. Correspondingly, a rain estimation network, a generative adversarial network and a refinement network are proposed to accomplish the three sub-tasks. The REN uses a dense network with three paths to estimate rain map in different perceptive fields. The generator is a residual network to remove rain streaks. The RFN is inspired by the super-resolution network and can make the de-raining result clearer and sharper. The experiments demonstrate that the proposed method outperforms state-of-the-art image de-raining methods on two synthetic datasets and real world rainy images. 

## Figures and Tables

**Figure 1 sensors-20-01591-f001:**

A de-raining example of proposed model on real world images.

**Figure 2 sensors-20-01591-f002:**
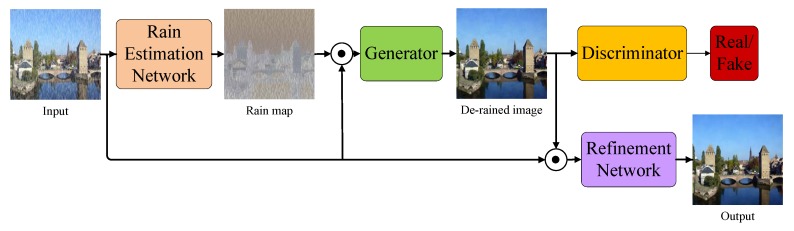
Overview of the proposed IGAN-SID method. REN aims to estimate the rain map. The generator, whose input is the concatenation of the rain map and the original rainy image, is designed for rain streak removal. The RFN, whose input is the concatenation of the de-raining result and the original rainy image, aims to restore the lost details in the de-raining result.

**Figure 3 sensors-20-01591-f003:**
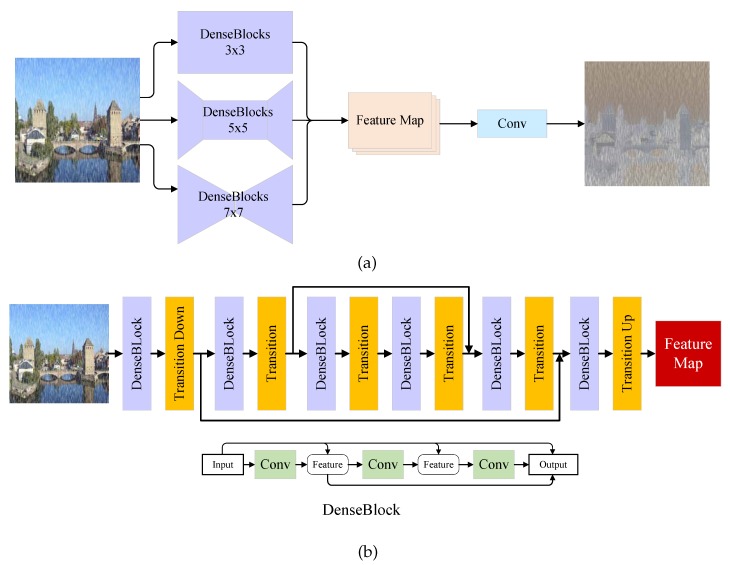
The architecture of Rain Estimation Network. (**a**) The overview of REN. It consists of three-stream DenseNet [23] with different kernel size, i.e., 3×3, 5×5 and 7×7. (**b**) The architecture of 5×5 DenseNet stream. The transition down means a stride-2 convolution layer with ReLU and IN. The transition up means a stride-2 deconvolution layer with ReLU and IN. The transition up means a stride-1 convolution layer with ReLU and IN. Each dense block has four convolution layers as shown.

**Figure 4 sensors-20-01591-f004:**
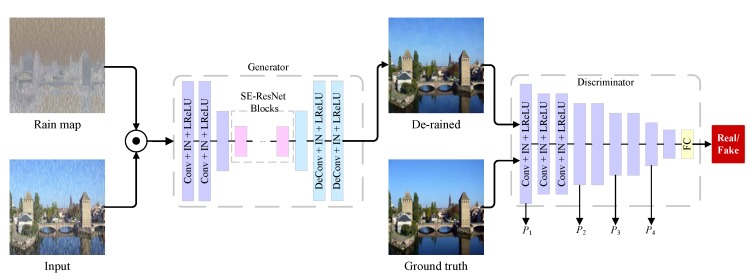
The architecture of proposed GAN model. The generator consists of stride convolution layers, residual blocks and stride de-convolution layers. The discriminator is a binary classification based on CNNs. Reproduced with permission from Li et al. [13], published by 2019 IEEE International Conference on Multimedia and Expo (ICME).

**Figure 5 sensors-20-01591-f005:**
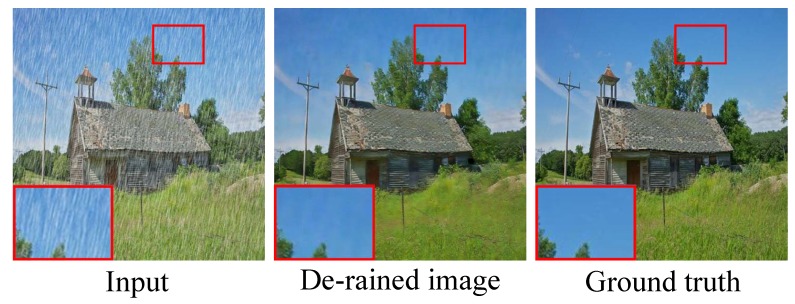
An example of rain residue after de-raining. It can be seen that the zoomed area still has rain residue which should be refined.

**Figure 6 sensors-20-01591-f006:**
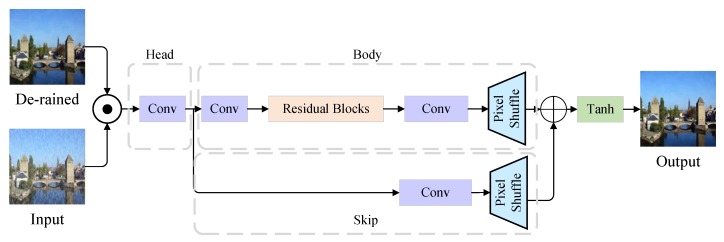
The architecture of Refine Network. It consists of head, body and skip parts.

**Figure 7 sensors-20-01591-f007:**

An example of comparison between batch normalization and instance normalization. From left to right: input, using batch normalization, using instance normalization, ground truth.Reproduced with permission from Li et al. [13], published by 2019 IEEE International Conference on Multimedia and Expo (ICME).

**Figure 8 sensors-20-01591-f008:**
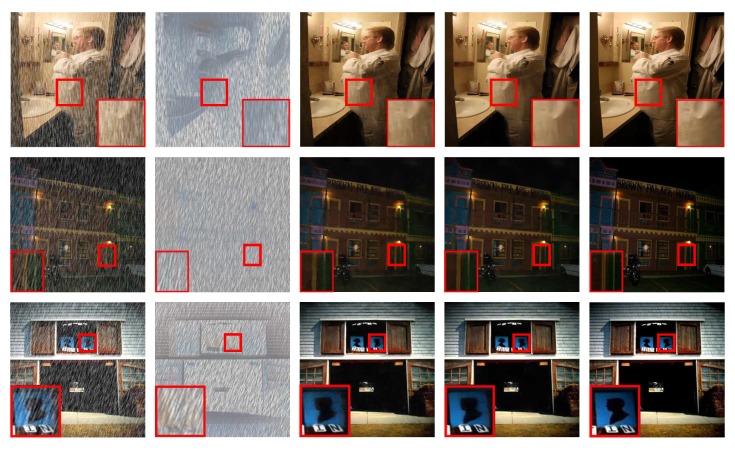
Several example of proposed model. From left to right are: input, rain map, output of generator, output of RFN, ground truth.

**Figure 9 sensors-20-01591-f009:**
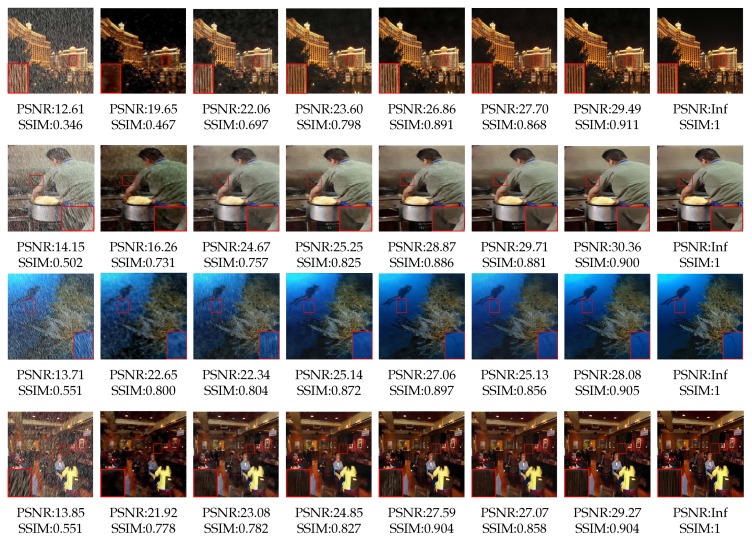
Visual comparison results on synthesic *DID-MDN dataset*. From left to right: Input, JORDER [8], DDN [7], DID-MDN [6], PReNet [16], GAN-SID [13], Ours, Ground Truth.

**Figure 10 sensors-20-01591-f010:**
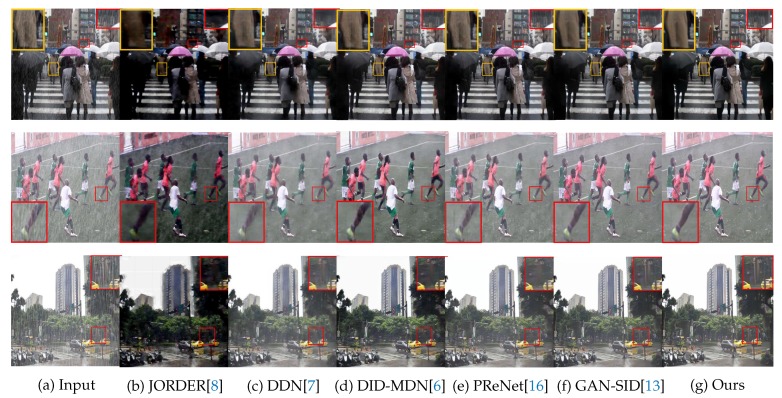
Visual comparison results on real-world images.

**Table 1 sensors-20-01591-t001:** Results of ablation experiments.

	w/o IN	w/o SE	w/o PL	GAN+REN	GAN+REN+RFN
PSNR	27.85	30.69	30.58	31.06	**32.27**
SSIM	0.880	0.927	0.925	0.934	**0.942**

**Table 2 sensors-20-01591-t002:** Parameters and time overhead comparison.

	GAN	GAN+REN	GAN+REN+RFN
Parameters (M)	8.475	8.666	9.862
Time (s)	0.029	0.031	0.037

**Table 3 sensors-20-01591-t003:** Quantitative results on *ID-CGAN dataset*.

	ID-CGAN [10]	PAN [11]	GAN-SID [13]	IGAN-SID (Ours)
PSNR	22.91	23.35	25.47	**25.84**
SSIM	0.820	0.830	0.856	**0.871**

**Table 4 sensors-20-01591-t004:** Quantitative comparison results on *DID-MDN dataset*.

	JORDER [8]	DDN [7]	JBO [2]	DID-MDN [6]	PReNet [16]	GAN-SID [13]	Ours
PSNR	24.32	27.33	23.05	27.95	31.12	30.97	**32.26**
SSIM	0.862	0.898	0.8522	0.909	0.939	0.931	**0.942**

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
