# Peer review of "Single Image De-Raining via Improved Generative Adversarial Netsâ€"

_sensors, 2020, doi:10.3390/s20061591_

Round 1
Reviewer 1 Report
This paper presents a generative adversarial network for image de-raining. It is an extension of the method already presented by the authors in:
S. Li, Y. Hou, H. Yue and Z. Guo, "Single Image De-Raining via Generative Adversarial Nets," 2019 IEEE International Conference on Multimedia and Expo (ICME), Shanghai, China, 2019, pp. 1192-1197.
I see an improvement over the previous work in terms of design of the network and PSNR/SSIM values. Moreover, each section of this new paper has been rewritten and presents novel information compared to their previous work.
However, the overall English of the paper is poor, which is quite annoying. I heavily recommend the authors to carefully proofread the paper or to review it by a native speaker in order to correct some mistakes such as:
· Lines 6-8: i.e. is overused
· Line 15: essential
· Paragraphs 35-38 and 39-43 are almost the same
· Line 206-207: "which was published" is repeated twice
...
Apart from that, I have an observation and some minor comments:
· Figure 2: I think that the Discriminator should be placed after the refinement network. If not then, why discriminating an image that is not refined?
Minor comments:
· References are not cited in order. Ref [22] should be reordered.
· Line 17: convolutional
· Line 18: GAN is defined in line 28, it should be defined here.
· Lines 27-28: Expression Y = R + C is not neccessary here. I would say something like this "the pixels of the image with rain consist of rain steaks and pixels of the clear image without rain."
· Figure 1. Instead of "Ours" I would say "Proposed method". Check the use of "Ours" along the paper.
· Line 46: GAN was already defined in line 28. Check this type of error since there are several acronyms redefined along the paper over and over.
· Please, separate references from words. E.g. DenseNet [13] instead of DenseNet[13]
· Table 2 should be placed after Table 1 and before subsection 4.3.2
· Figures are not correctly placed in the paper. For example, Fig.2 is located on page 3, but it is referenced at the end of page 4. Similarly, this issue occurs with the rest of the figures.
· The expression i.e. is overused along the paper.
Author Response
Thanks for your professional work. The valuable comments of the reviewers are very useful in significantly improving the quality of this paper.We have enclosed a point-by-point response to address all your comments in great detail. We hope that the revised manuscript can meet your expectation.
Comment 1: The overall English of the paper is poor, which is quite annoying. I heavily recommend the authors to carefully proofread the paper or to review it by a native speaker in order to correct some mistakes such as:
- Lines 6-8: i.e. is overused
Response: We have deleted too many uses of i.e. And we have rewritten the sentence as: These three models accomplish rain locating, rain removing, detail refining sub-tasks respectively.
- Line 15: essential
Response: As suggested, typo essential has been fixed.
- Paragraphs 35-38 and 39-43 are almost the same
Response: As suggested, we have deleted the following sentences:
1) lines 36-38: We extend GAN-SID [12] by decomposing the de-raining task into three specific tasks, i.e., rain locating, rain removing, and detail refining.
2) line 40: We propose a novel GAN-based image de-raining network called IGAN-SID.
And the following sentence has been modified as:
1) line 40: “we utilize the divide-and-conquer strategy.” has been rewritten as “the proposed model utilizes the divide-and-conquer strategy.”
- Line 206-207: "which was published" is repeated twice.
Response: As suggested, we have rewritten the sentence as: “They are DID-MDN dataset [6] and ID-CGAN dataset [10].”.
Thanks for your suggestions, we have thoroughly read the paper and corrected spelling and grammar errors, the others modifications have been listed in the last parts of this response and we have marked it in red color in revised manuscript.
Comment 2: Figure 2: I think that the Discriminator should be placed after the refinement network. If not then, why discriminating an image that is not refined?
Response: Thanks for your comment. As per our experiments, placing Discriminator after refinement network has limited performance improvement. We consider that the goal of the refinement network is to restore the details at pixel level, MSE is the best loss in this sense. However, the refinement network will be constrained by the GAN loss if it is placed before Discriminator. Since the goal of GAN loss is guide the generated images close to real-world images in terms of high-level semantic information rather than per-pixel information. So that it is difficult to refine the results completely at the pixel level if the Discriminator is placed after refinement network.
Comment 3 References are not cited in order. Ref [22] should be reordered.
Response: Appreciate a lot for comments. We have reordered the references.
Comment 4: Line 17: convolutional
Response: As suggested, typo convolutional is fixed.
Comment 5: Line 18: GAN is defined in line 28, it should be defined here.
Response: As suggested, GAN is defined in line 19 (previous line 18) in revised manuscript. And “(GAN)” in line 28 has been removed.
Comment 6: Lines 27-28: Expression Y = R + C is not necessary here. I would say something like this "the pixels of the image with rain consist of rain steaks and pixels of the clear image without rain."
Response: Thanks for your suggestion. The expression Y = R + C has been removed. As suggested, the expression is corrected to: “However, the pixels of rain image are superposition of rain streaks and pixels of rain-free background image.”
Comment 7: Figure 1. Instead of "Ours" I would say "Proposed method". Check the use of "Ours" along the paper.
Response: Appreciate for your suggestions. As suggested, we have made the following revisions:
- In figure 1, the sentence has been rewritten as “A de-raining example of proposed model on real world images.”
- In line 177, the sentence has been rewritten as “The generative adversarial objective function for proposed model can be expressed as”.
- In line 206-207, the sentence has been rewritten as “But the proposed model can directly estimate the distribution of rain streaks, so that these labels are not used in this work.”
- In line 247, “our” is removed.
- In line 257, the sentence has been rewritten as “the output of the proposed model”.
- In line 284, the use of our has been removed and the paragraph has been written in the revised manuscript.
Comment 8: Line 46: GAN was already defined in line 28. Check this type of error since there are several acronyms redefined along the paper over and over.
Response: Thanks for pointing this mistake. As suggested, we have made the revisions as:
- In line 44 (line 46 in previous version), line 91, line 112 and line 311, the definition of GAN is removed.
- In figure 2, line 112, line 122, line 310, the definition of REN is removed.
- In figure 2, line 112, line 189, line 311, the definition of RFN is removed.
- The definitions of JORDER, DDN, DID-MDN are removed which had been defined in section 2.
- The definitions of IN are removed which had been defined in section 1.
Comment 9: Please, separate references from words. E.g. DenseNet [13] instead of DenseNet[13]
Response: Thanks for pointing out this mistake. As suggested, all references have been separated in the revised manuscripts.
Comment 10: Table 2 should be placed after Table 1 and before subsection 4.3.2 Figures are not correctly placed in the paper. For example, Fig.2 is located on page 3, but it is referenced at the end of page 4. Similarly, this issue occurs with the rest of the figures.
Response: Thanks for suggestion. As suggested, we have rearranged the order of figures and tables to make them as reasonable as possible.
Comment 11: The expression i.e. is overused along the paper.
Response: Thanks for suggestion. We have made the following revisions:
- In lines 6-8, the uses of “e.” have been removed, and a sentence has been added as: These three models accomplish rain locating, rain removing, detail refining sub-tasks respectively.
- In lines 39-40, the sentence has been rewritten as “The de-raining task is decomposed into rain estimating, rain removing, and detail refining sub-tasks.”
- In line 42-43, the sentence has been rewritten as “The estimated rain location map can guide the followed de-raining generative network to generate better de-raining result.”
- In line 80, the sentence has been rewritten as “Different from other studies, DID-MDN can automatically estimate the rain-density label.”
- In line 174, the sentence has been rewritten as “Specifically, the generator aims to generate samples which is similar to real-world rain-free images.”
- In line 214, the sentence has been rewritten as “Specifically, update generator 5 times after updating discriminator once.”
- In line 219, the sentence has been rewritten as “For DID-MDN dataset, the image resolution is 512”
- In line 226, the sentence has been rewritten as “In other words, the samples will be affected by the others within same mini-batch.”
- In line 247, the sentence has been rewritten as “This is consistent with intuition that the rain streaks have slighter influence in high brightness area than low brightness area.”
- In line 267-291, the uses of e. have been removed and the paragraphs have been written in revised manuscript.
- In line 297, the sentence has been rewritten as “Correspondingly, a rain estimation network, a generative adversarial network and a refinement network are proposed to accomplish the three sub-tasks.”

Reviewer 2 Report
- Please explain abbreviation MSE.
- Based on which criteria and reasoning did you choose the values of parameters to be 5 and 10 in equations (8) and (10)?
- Please explain what programming language was used to perform experiments.
- Please give computer performances on which the experiments were executed.
- Please explain abbreviation w/o used on page 10.
- Please correct minor language errors like:
- “It play a role in single image de-raining task by modeling the importance of feature channels.”, page 6.
- “GAN loss function gives adversarial goals of generator and discriminator, i.e., the generator aims to generate samples which similar to real-world rain-free images.”, page 7.
- “After apply BN operation, the output features of neurons are normalized to mean value 0 and variance value 1 at batch scale.”, page 10.
- “IN is suitable for image de-raining task due to the fact that it just normalize data at instance scale.”, page 10.
- “It can be observed that REN can estimate the rain streaks effectively. In the area of high brightness, the estimated rain streaks are sparse or even not exists.”, page 11.
- “Another reason is that the Instance Normalization which used in proposed model is more beneficial for image de-raining task, as analyzed before.”, page 11.
- “Thus the proposed model could performance much better than other GAN-based models.”, page 12.
- “The results shows that take single image de-raining task as an image-to-image translation task is an idea that worth exploring.”, page 12.
- “To visually demonstrate the improvements which obtained by the proposed method, Fig.9 shows visual results from synthetic rainy images.”, page 12.
Author Response
Thanks for your professional work. The valuable comments of the reviewers are very useful in significantly improving the quality of this paper.We have enclosed a point-by-point response to address all your comments in great detail. We hope that the revised manuscript can meet your expectation.
Comment 1: Please explain abbreviation MSE
Response: Thanks for your comments. Mean-Square Error (MSE) is a per-pixel loss function. It can be described as L2 distance.
As suggested, we have added the explanation in section 3.4.3 as: Mean-Square Error (MSE) loss is applied in GAN model as per-pixel loss. … the MSE loss function can be described as L2 distance…
Comment 2: Based on which criteria and reasoning did you choose the values of parameters to be 5 and 10 in equations (8) and (10)?
Response: Thanks for comments. It is an important and common issue to select the network hyperparameters. Up to now, the hyperparameters of network are empirically set. We determined these hyperparameters through experiments.
Comment 3: Please explain what programming language was used to perform experiments
Response: Thanks for comment. All the experiments were performed on programming language python3 and deep-learning framework pytorch 0.4.0. The evaluations were performed on MATLAB 2017. As suggested, we have added the explanation in first paragraph of section 4.2 as: All experiments are carried on programming language python and deep-learning framework pytorch. The performance is evaluated on MATLAB.
Comment 4: Please give computer performances on which the experiments were executed
Response: Thanks for comment. We executed experiments with GPU Titan which has 12 GB memory, CPU i7-6700 and RAM 16G. As suggested, we have added the explanation in first paragraph of section 4.2 as: The experiments are executed on GPU Nvidia Titan which has 12GB memory, CPU i7-6700 and RAM 16G.
Comment 5: Please explain abbreviation w/o used on page 10.
Response: Appreciate for comments. w/o means the proposed model without one part and keep the other parts unchanged. For example, w/o IN means the proposed model without IN application. Instead, BN is applied. As suggested, we have added the modification in the first paragraph of section 4.3.1 as: Where w/o means the proposed model without one part and keep the other parts unchanged.
Comment 6: Please correct minor language errors like:
- “It play a role in single image de-raining task by modeling the importance of feature channels.”, page 6.
Response: “play” has been corrected to “plays” in line 143.
- “GAN loss function gives adversarial goals of generator and discriminator, i.e., the generator aims to generate samples which similar to real-world rain-free images.”, page 7.
Response: The sentence has been rewritten as “GAN loss function gives adversarial goals of generator and discriminator.” in line 173.
- “After apply BN operation, the output features of neurons are normalized to mean value 0 and variance value 1 at batch scale.”, page 10.
Response: “apply” has been corrected to “applying” in line 225.
- “IN is suitable for image de-raining task due to the fact that it just normalize data at instance scale.”, page 10.
Response: “normalize” has been corrected to “normalizes” in line 229.
- “It can be observed that REN can estimate the rain streaks effectively. In the area of high brightness, the estimated rain streaks are sparse or even not exists.”, page 11.
Response: We have rewritten the sentence as “In the area of high brightness, rain streaks cannot be estimated effectively.” in line 247
- “Another reason is that the Instance Normalization which used in proposed model is more beneficial for image de-raining task, as analyzed before.”, page 11.
Response: “is more beneficial for” has been corrected as “is more beneficial to” in line 258
- “Thus the proposed model could performance much better than other GAN-based models.”, page 12.
Response: The sentence has been corrected to “Thus the proposed model outperforms other GAN-based models.” in line 261.
- “The results shows that take single image de-raining task as an image-to-image translation task is an idea that worth exploring.”, page 12.
Response: “The results shows” has been corrected to “The results show” in line 265
- “To visually demonstrate the improvements which obtained by the proposed method, Fig.9 shows visual results from synthetic rainy images.”, page 12.
Response: The paragraph has been rewritten in revised manuscript.
Thanks for your suggestions, we have thoroughly read the paper and corrected spelling and grammar errors, the others modifications have been listed in the last parts of this response and we have marked it in red color in revised manuscript.
